# Do Convnets Learn Correspondence?

**Jonathan Long**　　　**Ning Zhang**　　　**Trevor Darrell**
University of California – Berkeley
`{jonlong, nzhang, trevor}@cs.berkeley.edu`

## Abstract

Convolutional neural nets (convnets) trained from massive labeled datasets [1] have substantially improved the state-of-the-art in image classification [2] and object detection [3]. However, visual understanding requires establishing correspondence on a finer level than object category. Given their large pooling regions and training from whole-image labels, it is not clear that convnets derive their success from an accurate correspondence model which could be used for precise localization. In this paper, we study the effectiveness of convnet activation features for tasks requiring correspondence. We present evidence that convnet features localize at a much finer scale than their receptive field sizes, that they can be used to perform intraclass alignment as well as conventional hand-engineered features, and that they outperform conventional features in keypoint prediction on objects from PASCAL VOC 2011 [4].

## 1   Introduction

Recent advances in convolutional neural nets [2] dramatically improved the state-of-the-art in image classification. Despite the magnitude of these results, many doubted [5] that the resulting features had the spatial specificity necessary for localization; after all, whole image classification can rely on context cues and overly large pooling regions to get the job done. For coarse localization, such doubts were alleviated by record breaking results extending the same features to detection on PASCAL [3].

Now, the same questions loom on a finer scale. Are the modern convnets that excel at classification and detection also able to find precise correspondences between object parts? Or do large receptive fields mean that correspondence is effectively pooled away, making this a task better suited for hand-engineered features?

In this paper, we provide evidence that convnet features perform at least as well as conventional ones, even in the regime of point-to-point correspondence, and we show considerable performance improvement in certain settings, including category-level keypoint prediction.

### 1.1   Related work

**Image alignment**   Image alignment is a key step in many computer vision tasks, including face verification, motion analysis, stereo matching, and object recognition. Alignment results in correspondence across different images by removing intraclass variability and canonicalizing pose. Alignment methods exist on a supervision spectrum from requiring manually labeled fiducial points or landmarks, to requiring class labels, to fully unsupervised joint alignment and clustering models. Congealing [6] is an unsupervised joint alignment method based on an entropy objective. Deep congealing [7] builds on this idea by replacing hand-engineered features with unsupervised feature learning from multiple resolutions. Inspired by optical flow, SIFT flow [8] matches densely sampled SIFT features for correspondence and has been applied to motion prediction and motion transfer. In Section 3, we apply SIFT flow using deep features for aligning different instances of the same class.

**Keypoint localization**   Semantic parts carry important information for object recognition, object detection, and pose estimation. In particular, fine-grained categorization, the subject of many recent works, depends strongly on part localization [9, 10]. Large pose and appearance variation across examples make part localization for generic object categories a challenging task.

Most of the existing works on part localization or keypoint prediction focus on either facial landmark localization [11] or human pose estimation. Human pose estimation has been approached using tree structured methods to model the spatial relationships between parts [12, 13, 14], and also using poselets [15] as an intermediate step to localize human keypoints [16, 17]. Tree structured models and poselets may struggle when applied to generic objects with large articulated deformations and wide shape variance.

**Deep learning**   Convolutional neural networks have gained much recent attention due to their success in image classification [2]. Convnets trained with backpropagation were initially succesful in digit recognition [18] and OCR [19]. The feature representations learned from large data sets have been found to generalize well to other image classification tasks [20] and even to object detection [3, 21]. Recently, Toshev et al. [22] trained a cascade of regression-based convnets for human pose estimation and Jain et al. [23] combine a weak spatial model with deep learning methods.

The latter work trains multiple small, independent convnets on $64 \times 64$ patches for binary body-part detection. In contrast, we employ a powerful pretained ImageNet model that shares mid-elvel feature representations among all parts in Section 5.

Several recent works have attempted to analyze and explain this overwhelming success. Zeiler and Fergus [24] provide several heuristic visualizations suggesting coarse localization ability. Szegedy et al. [25] show counterintuitive properties of the convnet representation, and suggest that individual feature channels may not be more semantically meaningful than other bases in feature space. A concurrent work [26] compares convnet features with SIFT in a standard descriptor matching task. This work illuminates and extends that comparison by providing visual analysis and by moving beyond single instance matching to intraclass correspondence and keypoint prediction.

## 1.2   Preliminaries

We perform experiments using a network architecture almost identical[1] to that popularized by Krizhevsky et al. [2] and trained for classification using the 1.2 million images of the ILSVRC 2012 challenge dataset [1]. All experiments are implemented using `caffe` [27], and our network is the publicly available `caffe` reference model. We use the activations of each layer as features, referred to as `conv`$n$, `pool`$n$, or `fc`$n$ for the $n$th convolutional, pooling, or fully connected layer, respectively. We will use the term *receptive field*, abbreviated rf, to refer to the set of input pixels that are path-connected to a particular unit in the convnet.

## 2   Feature visualization

In this section and Figures 1 and 2, we provide a novel visual investigation of the effective pooling regions of convnet features.

In Figure 1, we perform a nonparametric reconstruction of images from features in the spirit of HOGgles [28]. Rather than paired dictionary learning, however, we simply replace patches with averages of their top-$k$ nearest neighbors in a convnet feature space. To do so, we first compute all features at a particular layer, resulting in an 2d grid of feature vectors. We associate each feature vector with a patch in the

Table 1: Convnet receptive field sizes and strides, for an input of size $227 \times 227$.

| layer | rf size | stride |
|-------|---------|--------|
| `conv1` | $11 \times 11$ | $4 \times 4$ |
| `conv2` | $51 \times 51$ | $8 \times 8$ |
| `conv3` | $99 \times 99$ | $16 \times 16$ |
| `conv4` | $131 \times 131$ | $16 \times 16$ |
| `conv5` | $163 \times 163$ | $16 \times 16$ |
| `pool5` | $195 \times 195$ | $32 \times 32$ |

original image at the center of the corresponding receptive field and with size equal to the receptive field stride. (Note that the strides of the receptive fields are much smaller than the receptive fields

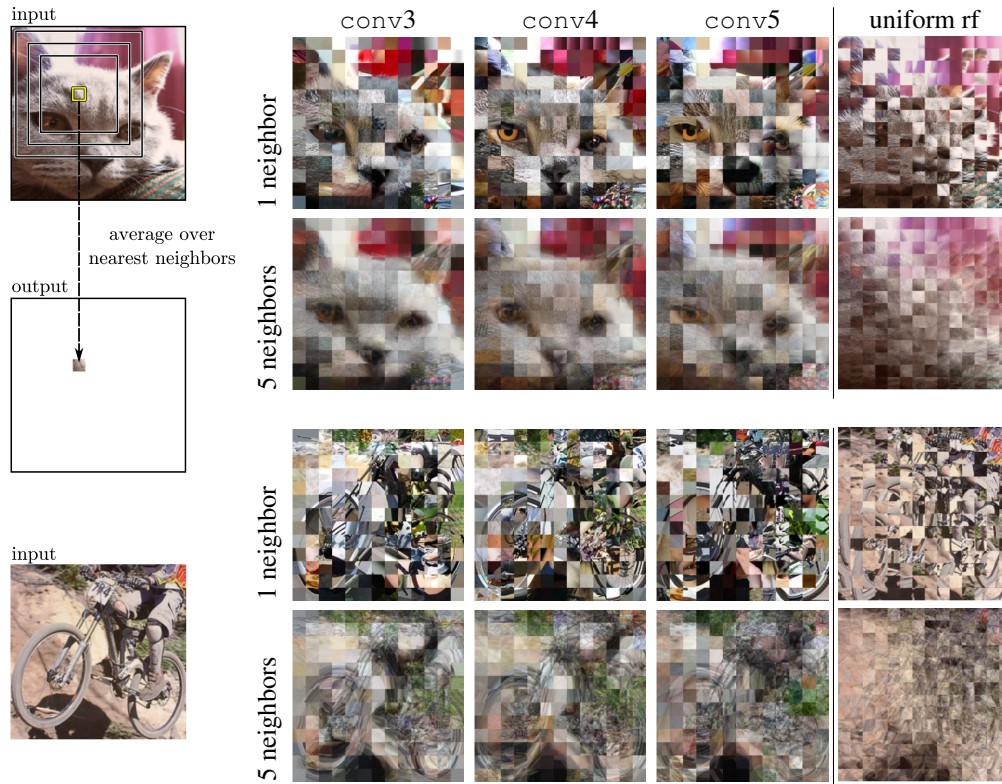

Figure 1: Even though they have large receptive fields, convnet features carry local information at a finer scale. Upper left: given an input image, we replaced $16 \times 16$ patches with averages over 1 or 5 nearest neighbor patches, computed using convnet features centered at those patches. The yellow square illustrates one input patch, and the black squares show the corresponding rfs for the three layers shown. Right: Notice that the features retrieve reasonable matches for the centers of their receptive fields, even though those rfs extend over large regions of the source image. In the "uniform rf" column, we show the best that could be expected if convnet features discarded all spatial information within their rfs, by choosing input patches uniformly at random from conv3-sized neighborhoods. (Best viewed electronically.)

themselves, which overlap. Refer to Table 1 above for specific numbers.) We replace each such patch with an average over $k$ nearest neighbor patches using a database of features densely computed on the images of PASCAL VOC 2011. Our database contains at least one million patches for every layer. Features are matched by cosine similarity.

Even though the feature rfs cover large regions of the source images, the specific resemblance of the resulting images shows that information is not spread uniformly throughout those regions. Notable features (e.g., the tires of the bicycle and the facial features of the cat) are replaced in their corresponding locations. Also note that replacement appears to become more semantic and less visually specific as the layer deepens: the eyes and nose of the cat get replaced with differently colored or shaped eyes and noses, and the fur gets replaced with various animal furs, with the diversity increasing with layer number.

Figure 2 gives a feature-centric rather than image-centric view of feature locality. For each column, we first pick a random seed feature vector (computed from a PASCAL image), and find $k$ nearest neighbor features, again by cosine similarity. Instead of averaging only the centers, we average the entire receptive fields of the neighbors. The resulting images show that similar features tend to respond to similar colors specifically in the centers of their receptive fields.

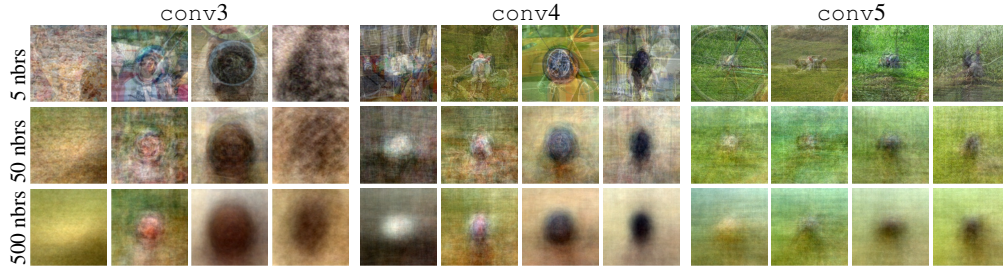

Figure 2: Similar convnet features tend to have similar receptive field centers. Starting from a randomly selected seed patch occupying one rf in `conv3`, 4, or 5, we find the nearest $k$ neighbor features computed on a database of natural images, and average together the corresponding receptive fields. The contrast of each image has been expanded after averaging. (Note that since each layer is computed with a stride of 16, there is an upper bound on the quality of alignment that can be witnessed here.)

## 3   Intraclass alignment

We conjecture that category learning implicitly aligns instances by pooling over a discriminative mid-level representation. If this is true, then such features should be useful for post-hoc alignment in a similar fashion to conventional features. To test this, we use convnet features for the task of aligning different instances of the same class. We approach this difficult task in the style of SIFT flow [8]: we retrieve near neighbors using a coarse similarity measure, and then compute dense correspondences on which we impose an MRF smoothness prior which finally allows all images to be warped into alignment.

Nearest neighbors are computed using `fc7` features. Since we are specifically testing the quality of alignment, we use the same nearest neighbors for convnet or conventional features, and we compute both types of features at the same locations, the grid of convnet rf centers in the response to a single image.

Alignment is determined by solving an MRF formulated on this grid of feature locations. Let $p$ be a point on this grid, let $f_s(p)$ be the feature vector of the source image at that point, and let $f_t(p)$ be the feature vector of the target image at that point. For each feature grid location $p$ of the source image, there is a vector $w(p)$ giving the displacement of the corresponding feature in the target image. We use the energy function

$$E(w) = \sum_p \|f_s(p) - f_t(p + w(p))\|_2 + \beta \sum_{(p,q)\in\mathcal{E}} \|w(p) - w(q)\|_2^2,$$

where $\mathcal{E}$ are the edges of a 4-neighborhood graph and $\beta$ is the regularization parameter. Optimization is performed using belief propagation, with the techniques suggested in [29]. Message passing is performed efficiently using the squared Euclidean distance transform [30]. (Unlike the $L_1$ regularization originally used by SIFT flow [8], this formulation maintains rotational invariance of $w$.)

Based on its performance in the next section, we use `conv4` as our convnet feature, and SIFT with descriptor radius 20 as our conventional feature. From validation experiments, we set $\beta = 3 \cdot 10^{-3}$ for both `conv4` and SIFT features (which have a similar scale).

Given the alignment field $w$, we warp target to source using bivariate spline interpolation (implemented in SciPy [31]). Figure 3 gives examples of alignment quality for a few different seed images, using both SIFT and convnet features. We show five warped nearest neighbors as well as keypoints transferred from those neighbors.

We quantitatively assess the alignment by measuring the accuracy of predicted keypoints. To obtain good predictions, we warp 25 nearest neighbors for each target image, and order them from smallest to greatest deformation energy (we found this method to outperform ordering using the data term). We take the predicted keypoints to be the median points (coordinate-wise) of the top five aligned keypoints according to this ordering.

We assess correctness using mean PCK [32]. We consider a ground truth keypoint to be correctly predicted if the prediction lies within a Euclidean distance of $\alpha$ times the maximum of the bounding

target image             five nearest neighbors

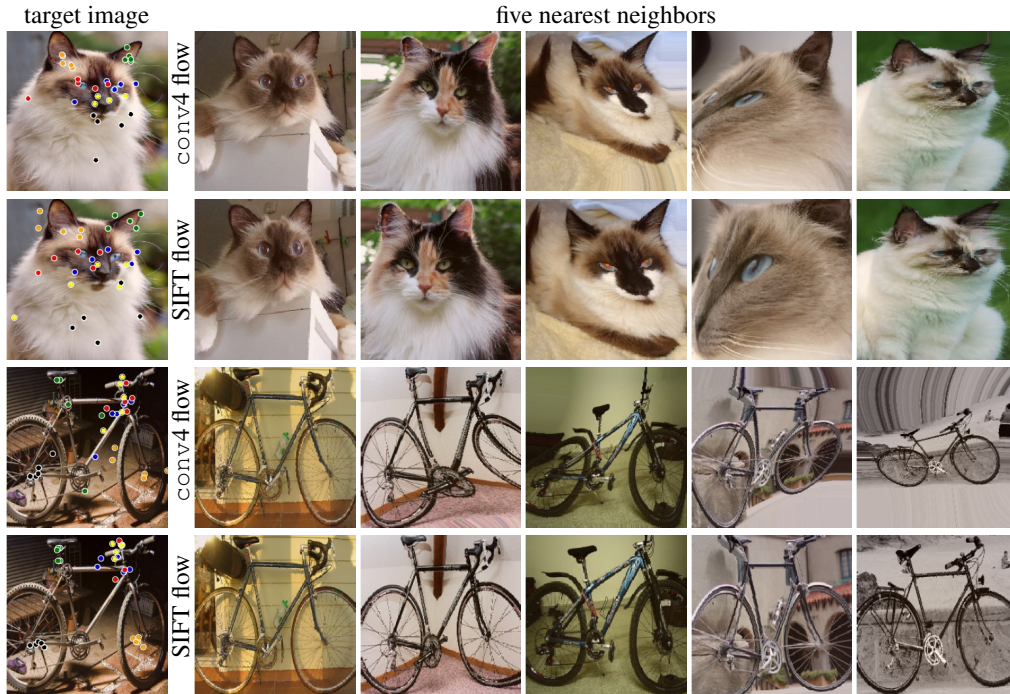

Figure 3: Convnet features can bring different instances of the same class into good alignment at least as well (on average) as traditional features. For each target image (left column), we show warped versions of five nearest neighbor images aligned with `conv4` flow (first row), and warped versions aligned with SIFT flow [8] (second row). Keypoints from the warped images are shown copied to the target image. The cat shows a case where convnet features perform better, while the bicycle shows a case where SIFT features perform better. (Note that each instance is warped to a square bounding box before alignment. Best viewed in color.)

Table 2: Keypoint transfer accuracy using convnet flow, SIFT flow, and simple copying from nearest neighbors. Accuracy (PCK) is shown per category using $\alpha = 0.1$ (see text) and means are also shown for the stricter values $\alpha = 0.05$ and $0.025$. On average, convnet flow performs as well as SIFT flow, and performs a bit better for stricter tolerances.

| | aero | bike | bird | boat | bttl | bus | car | cat | chair | cow | table | dog | horse | mbike | prsn | plant | sheep | sofa | train | tv | mean |
|---|---|---|---|---|---|---|---|---|---|---|---|---|---|---|---|---|---|---|---|---|---|
| conv4 flow | 28.2 | 34.1 | 20.4 | 17.1 | 50.6 | 36.7 | 20.9 | 19.6 | 15.7 | 25.4 | 12.7 | 18.7 | 25.9 | 23.1 | 21.4 | 40.2 | 21.1 | 14.5 | 18.3 | 33.3 | 24.9 |
| SIFT flow | 27.6 | 30.8 | 19.9 | 17.5 | 49.4 | 36.4 | 20.7 | 16.0 | 16.1 | 25.0 | 16.1 | 16.3 | 27.7 | 28.3 | 20.2 | 36.4 | 20.5 | 17.2 | 19.9 | 32.9 | 24.7 |
| NN transfer | 18.3 | 24.8 | 14.5 | 15.4 | 48.1 | 27.6 | 16.0 | 11.1 | 12.0 | 16.8 | 15.7 | 12.7 | 20.2 | 18.5 | 18.7 | 33.4 | 14.0 | 15.5 | 14.6 | 30.0 | 19.9 |

| mean | $\alpha = 0.1$ | $\alpha = 0.05$ | $\alpha = 0.025$ |
|---|---|---|---|
| conv4 flow | 24.9 | 11.8 | 4.08 |
| SIFT flow | 24.7 | 10.9 | 3.55 |
| NN transfer | 19.9 | 7.8 | 2.35 |

box width and height, picking some $\alpha \in [0, 1]$. We compute the overall accuracy for each type of keypoint, and report the average over keypoint types. We do not penalize predicted keypoints that are not visible in the target image.

Results are given in Table 2. We show per category results using $\alpha = 0.1$, and mean results for $\alpha = 0.1$, $0.05$, and $0.025$. Indeed, convnet learned features are at least as capable as SIFT at alignment, and better than might have been expected given the size of their receptive fields.

## 4 Keypoint classification

In this section, we specifically address the ability of convnet features to understand semantic information at the scale of parts. As an initial test, we consider the task of *keypoint classification*: given an image and the coordinates of a keypoint on that image, can we train a classifier to label the keypoint?

Table 3: Keypoint classification accuracies, in percent, on the twenty categories of PASCAL 2011 val, trained with SIFT or convnet features. The best SIFT and convnet scores are bolded in each category.

|  |  | aero | bike | bird | boat | bttl | bus | car | cat | chair | cow | table | dog | horse | mbike | prsn | plant | sheep | sofa | train | tv | mean |
|---|---|------|------|------|------|------|-----|-----|-----|-------|-----|-------|-----|-------|-------|------|-------|-------|------|-------|-----|------|
| SIFT | 10 | 36 | 42 | 36 | 32 | 67 | 64 | 40 | 37 | 33 | 37 | 60 | 34 | 39 | 38 | 29 | 63 | 37 | 42 | 64 | 75 | 45 |
| (radius) | 20 | **37** | 50 | **39** | 35 | 74 | 67 | **47** | **40** | 36 | **43** | 68 | **38** | **42** | 48 | **33** | **70** | **44** | **52** | 68 | 77 | 50 |
|  | 40 | 35 | **54** | 37 | 41 | **76** | **68** | **47** | 37 | 39 | 40 | 69 | 36 | **42** | 49 | 32 | 69 | 39 | **52** | **74** | **78** | 51 |
|  | 80 | 33 | 43 | 37 | **42** | 75 | 66 | 42 | 30 | **43** | 36 | **70** | 31 | 36 | **51** | 27 | **70** | 35 | 49 | 69 | 77 | 48 |
|  | 160 | 27 | 36 | 34 | 38 | 72 | 59 | 35 | 25 | 39 | 30 | 67 | 27 | 32 | 46 | 25 | **70** | 29 | 48 | 66 | 76 | 44 |
| conv | 1 | 16 | 14 | 15 | 19 | 20 | 29 | 15 | 22 | 16 | 17 | 29 | 17 | 14 | 16 | 15 | 33 | 18 | 12 | 27 | 29 | 20 |
| (layer) | 2 | 37 | 43 | 40 | 35 | 69 | 63 | 38 | 44 | 35 | 40 | 61 | 38 | 40 | 44 | 34 | 65 | 39 | 41 | 63 | 72 | 47 |
|  | 3 | 42 | 50 | 46 | 41 | 76 | 69 | **46** | 52 | 39 | 45 | 64 | 47 | 48 | 52 | 40 | 74 | 46 | 50 | 71 | **77** | 54 |
|  | 4 | **44** | **53** | **49** | **42** | **78** | **70** | 45 | **55** | **41** | **48** | **68** | **51** | **51** | **53** | **41** | **76** | **49** | **52** | **73** | 76 | **56** |
|  | 5 | **44** | 51 | **49** | 41 | 77 | 68 | 44 | 53 | 39 | 45 | 63 | 50 | 49 | 52 | 39 | 73 | 47 | 47 | 71 | 75 | 54 |

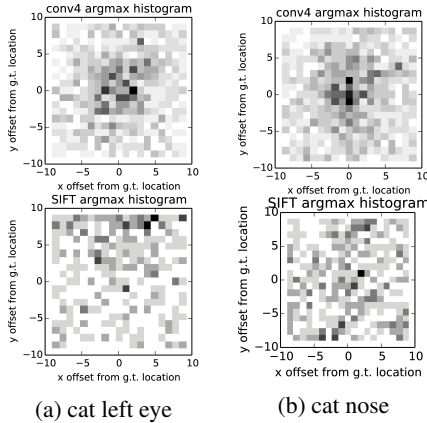

(a) cat left eye          (b) cat nose

Figure 4: Convnet features show fine localization ability, even beyond their stride and in cases where SIFT features do not perform as well. Each plot is a 2D histogram of the locations of the maximum responses of a classifer in a 21 by 21 pixel rectangle taken around a ground truth keypoint.

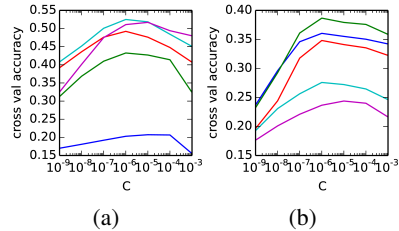

(a)          (b)

Figure 5: Cross validation scores for cat keypoint classification as a function of the SVM parameter $C$. In (a), we plot mean accuracy against $C$ for five different convnet features; in (b) we plot the same for SIFT features of different sizes. We use $C = 10^{-6}$ for all experiments in Table 3.

For this task we use keypoint data [15] on the twenty classes of PASCAL VOC 2011 [4]. We extract features at each keypoint using SIFT [33] and using the column of each convnet layer whose rf center lies closest to the keypoint. (Note that the SIFT features will be more precisely placed as a result of this approximation.) We trained one-vs-all linear SVMs on the train set using SIFT at five different radii and each of the five convolutional layer activations as features (in general, we found pooling and normalization layers to have lower performance). We set the SVM parameter $C = 10^{-6}$ for all experiments based on five-fold cross validation on the training set (see Figure 5).

Table 3 gives the resulting accuracies on the val set. We find features from convnet layers consistently perform at least as well as and often better than SIFT at this task, with the highest performance coming from layers conv4 and conv5. Note that we are specifically testing convnet features trained only for classification; the same net could be expected to achieve even higher performance if trained for this task.

Finally, we study the precise location understanding of our classifiers by computing their responses with a single-pixel stride around ground truth keypoint locations. For two example keypoints (cat left eye and nose), we histogram the locations of the maximum responses within a 21 pixel by 21 pixel rectangle around the keypoint, shown in Figure 4. We do not include maximum responses that lie on the boundary of this rectangle. While the SIFT classifiers do not seem to be sensitive to the precise locations of the keypoints, in many cases the convnet ones seem to be capable of localization finer than their strides, not just their receptive field sizes. This observation motivates our final experiments to consider detection-based localization performance.

# 5 Keypoint prediction

We have seen that despite their large receptive field sizes, convnets work as well as the hand-engineered feature SIFT for alignment and slightly better than SIFT for keypoint classification. Keypoint prediction provides a natural follow-up test. As in Section 3, we use keypoint annotations from PASCAL VOC 2011, and we assume a ground truth bounding box.

Inspired in part by [3, 34, 23], we train sliding window part detectors to predict keypoint locations independently. R-CNN [3] and OverFeat [34] have both demonstrated the effectiveness of deep convolutional networks on the generic object detection task. However, neither of them have investigated the application of CNNs for keypoint prediction.[2] R-CNN starts from bottom-up region proposal [35], which tends to overlook the signal from small parts. OverFeat, on the other hand, combines convnets trained for classification and for regression and runs in multi-scale sliding window fashion.

We rescale each bounding box to $500 \times 500$ and compute `conv5` (with a stride of 16 pixels). Each cell of `conv5` contains one 256-dimensional descriptor. We concatenate `conv5` descriptors from a local region of $3 \times 3$ cells, giving an overall receptive field size of $195 \times 195$ and feature dimension of 2304. For each keypoint, we train a linear SVM with hard negative mining. We consider the ten closest features to each ground truth keypoint as positive examples, and all the features whose rfs do not contain the keypoint as negative examples. We also train using dense SIFT descriptors for comparison. We compute SIFT on a grid of stride eight and bin size of eight using VLFeat [36]. For SIFT, we consider features within twice the bin size from the ground truth keypoint to be positives, while samples that are at least four times the bin size away are negatives.

We augment our SVM detectors with a spherical Gaussian prior over candidate locations constructed by nearest neighbor matching. The mean of each Gaussian is taken to be the location of the keypoint in the nearest neighbor in the training set found using cosine similarity on `pool5` features, and we use a fixed standard deviation of 22 pixels. Let $s(X_i)$ be the output score of our local detector for keypoint $X_i$, and let $p(X_i)$ be the prior score. We combine these to yield a final score $f(X_i) = s(X_i)^{1-\eta} p(X_i)^{\eta}$, where $\eta \in [0, 1]$ is a tradeoff parameter. In our experiments, we set $\eta = 0.1$ by cross validation. At test time, we predict the keypoint location as the highest scoring candidate over all feature locations.

We evaluate the predicted keypoints using the measure PCK introduced in Section 3, taking $\alpha = 0.1$. A predicted keypoint is defined as correct if the distance between it and the ground truth keypoint is less than $\alpha \cdot \max(h, w)$ where $h$ and $w$ are the height and width of the bounding box. The results using `conv5` and SIFT with and without the prior are shown in Table 4. From the table, we can see that local part detectors trained on the `conv5` feature outperform SIFT by a large margin and that prior information is helpful in both cases. To our knowledge, these are the first keypoint prediction results reported on this dataset. We show example results from five different categories in Figure 6. Each set consists of rescaled bounding box images with ground truth keypoint annotations and predicted keypoints using SIFT and `conv5` features, where each color corresponds to one keypoint. As the figure shows, `conv5` outperforms SIFT, often managing satisfactory outputs despite the challenge of this task. A small offset can be noticed for some keypoints like eyes and noses, likely due to the limited stride of our scanning windows. A final regression or finer stride could mitigate this issue.

# 6 Conclusion

Through visualization, alignment, and keypoint prediction, we have studied the ability of the intermediate features implicitly learned in a state-of-the-art convnet classifier to understand specific, local correspondence. Despite their large receptive fields and weak label training, we have found in all cases that convnet features are at least as useful (and sometimes considerably more useful) than conventional ones for extracting local visual information.

**Acknowledgements** This work was supported in part by DARPA's MSEE and SMISC programs, by NSF awards IIS-1427425, IIS-1212798, and IIS-1116411, and by support from Toyota.

Table 4: Keypoint prediction results on PASCAL VOC 2011. The numbers give average accuracy of keypoint prediction using the criterion described in Section 3, PCK with $\alpha = 0.1$.

| | aero | bike | bird | boat | bttl | bus | car | cat | chair | cow | table | dog | horse | mbike | prsn | plant | sheep | sofa | train | tv | mean |
|---|---|---|---|---|---|---|---|---|---|---|---|---|---|---|---|---|---|---|---|---|---|
| SIFT | 17.9 | 16.5 | 15.3 | 15.6 | 25.7 | 21.7 | 22.0 | 12.6 | 11.3 | 7.6 | 6.5 | 12.5 | 18.3 | 15.1 | 15.9 | 21.3 | 14.7 | 15.1 | 9.2 | 19.9 | 15.7 |
| SIFT+prior | 33.5 | 36.9 | 22.7 | 23.1 | 44.0 | 42.6 | 39.3 | 22.1 | 18.5 | 23.5 | 11.2 | 20.6 | 32.2 | 33.9 | 26.7 | 30.6 | 25.7 | 26.5 | 21.9 | 32.4 | 28.4 |
| conv5 | 38.5 | 37.6 | 29.6 | 25.3 | 54.5 | 52.1 | 28.6 | 31.5 | 8.9 | 30.5 | 24.1 | 23.7 | 35.8 | 29.9 | 39.3 | 38.2 | 30.5 | 24.5 | 41.5 | 42.0 | 33.3 |
| conv5+prior | **50.9** | **48.8** | **35.1** | **32.5** | **66.1** | **62.0** | **45.7** | **34.2** | **21.4** | **41.1** | **27.2** | **29.3** | **46.8** | **45.6** | **47.1** | **42.5** | **38.8** | **37.6** | **50.7** | **45.6** | **42.5** |

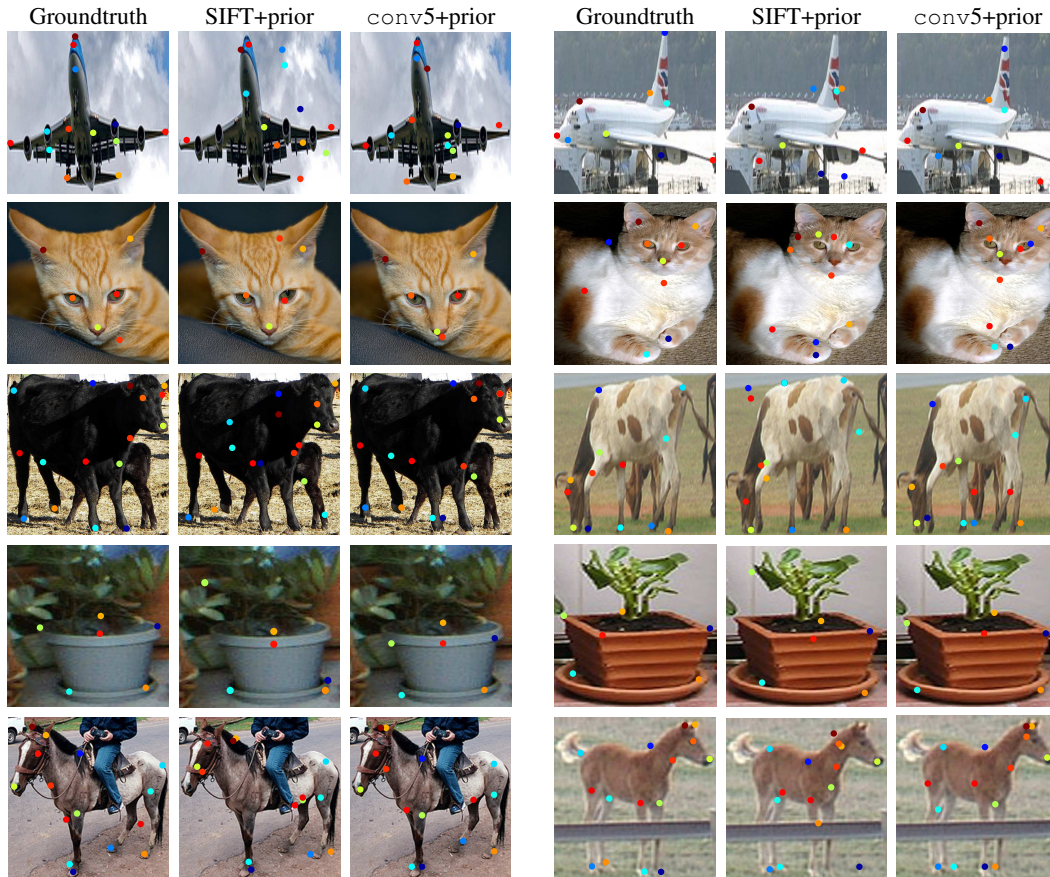

Figure 6: Examples of keypoint prediction on five classes of the PASCAL dataset: aeroplane, cat, cow, potted plant, and horse. Each keypoint is associated with one color. The first column is the ground truth annotation, the second column is the prediction result of SIFT+prior and the third column is conv5+prior. (Best viewed in color).

## Footnotes

[1]Ours reverses the order of the response normalization and pooling layers.

[2]But see works cited in Section 1.1 regarding keypoint localization.

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
