[Reviews · NeurIPS 2014]

Submitted by Assigned_Reviewer_8

In this paper the authors study the ability of convolutional networks to localize things precisely. The hypothesis is that convolutional networks are able to extract precise location information despite the coarseness of the features/receptive field sizes. The paper presents an interesting new way to visualize the features of the pooling regions in a conv-net, designs an experiment for perfoming intra-class alignment, and studies keypoint classification using conv-net features.

Quality: generally, the paper is sound and the experiments seem well-designed. Where the paper is lacking is a coherent message that connects the 3 experiments/visualizations -- they seem strung together somewhat haphazardly and I am not entirely certain what the main message is, besides that convolutional feature maps at higher layers can be used to localize/classify things precisely.

Clarity: mostly clearly written. I do find that the presentation of the experimental setups is very terse and reference-laden, at the expense of detailing the setup and the design choices made. There are numerous hyper-parameters (number of neighbords, thresholds etc) whose influence is not at all obvious wrt to the final numbers. In whatever version this submission will have next, I encourage the authors to spend more time detailing the experiments.

Originality: most of the ideas exposed in the submission seem new and interesting.

Significance: understanding how exactly convolutional networks are able to learn to detect and classify, and how the intermediate layers can be used for detection-like tasks.

Given the recent successes of using deep conv-nets for detection/classification tasks, such work is welcome, since it sheds some light into what the nets learn.

The feature visualization is interesting, but ultimately I am not sure how practical it is. The keypoint classification results seem strong, but the authors give conspicuously few hypotheses for the mechanisms why the conv-net features do as well as they do.
Summary: A few new ways to explore convolutional network feature maps and utilize them for alignment and keypoint classification. Generally, an interesting piece of work with good experimental results, but somewhat lacking in discussion.

Submitted by Assigned_Reviewer_29

The paper systematically addresses the following question: 'Are the modern convnets that excel at classification and detection also able to find correspondences between parts?' The authors do this through feature visualisation and experiments on intraclass alignment, keypoint classification and keypoint prediction. The findings are surprising: convnet features are at least as useful (and sometimes considerably more useful) than hand engineered features for the same tasks.

The question is original, the exposition is clear and the paper is a pleasure to read. This type of research takes us one step closer towards understanding why deep learning methods work as well as they do. I believe it will be of significant value to the NIPS community.

One criticism of the paper is that it does not contain much in the way of novel machine learning research, and it does not attempt to use its findings to advance the state-of-the-art.
Summary: The paper poses a clear question and explores it systematically through a series of carefully constructed experiments. The results are surprising and enhance our understanding of convnets.

Submitted by Assigned_Reviewer_44

The paper analyzes the degree to which features in CNNs identify object or part locations in images, even though they may not be trained for localization. There are three sets of tests. The first is to find nearest neighbors for patches based on their CNN activations; the second warps an image to its nearest neighbor patches, and the third transfers keypoints. The results of these experiments indicate that CNN features could be a replacement for SIFT in computer vision tasks.

Overall, the topic and results here generally justify publication of the paper. The paper addresses a very interesting and relevant question that could definitely have impact on the vision literature.

There are two major problems with the paper.

The first problem is that most of the results are very unconvincing as they are presented.
* The results in Figure 1 are not trustworthy. It is easy for a human viewer to be mislead by such a figure, i.e., to infer that correspondences are correct even when they are not, like in a Photomosaic (http://www.photomosaic.com/). More needs to be done to verify that the claimed correctness of correspondences is actually correct, like showing a visualization of class/part labels. Also, it doesn't help that the resolution of this figure is terrible; it is blurry even at print resolution.
* I do not understand the results in Figure 2 at all. It is unclear what is meant by receptive field here: I would assume it's the domain of the filters, but pixels are shown here.
* The results in Figure 3 are unpersuasive and ambiguous, but the scores in Table 2 seem fine to support the point that the results are comparable to SIFT (though not better).
* The results in Figure 6 and Table 4 are persuasive to me, but I'm not an expert in keypoint prediction, so I can't really judge the significance of this result or the experimental setup.

Second, the paper is not clearly written; it reads like a class project report, in which many terms and acronyms are used without definition. In many cases one can make educated guesses (and being familiar with the literature and the Caffe implementation helps), but one should not have to guess what is in the authors' heads. Examples: it is never defined how CNNs produce "features" (presumably the vector of filter outputs); the acronym "rfs" is never defined, nor is the receptive field that it refers to; in Section 3, the problem statement and inputs and outputs are never defined, nor is the source or target image. It's not actually clear to me what is meant by receptive field (see above).

One other comment: an impact of this paper is that it might encourage vision researchers to replace SIFT with CNN features. It would be helpful to hear about the practicalities of doing so, in particular the memory and computation time requirements of doing so.
Summary: Useful paper to help understand localization in CNNs, and how their convolutional layers might serve as features for general computer vision tasks. Useful work. However, results are sometimes ambiguously presented, and writing is inadequate to understand what the authors did and why they these results show what they think they show.
Author Feedback
Author rebuttal: We thank the reviewers for their helpful comments. We will address the concerns
of each in turn.

Our own view of the paper is much like that of Reviewer 29. In particular, our
aim with this work is precisely to get "one step closer towards understanding
why deep learning methods work as well as they do", which we think is a major
open question that will require a preponderance of evidence to address. We think
that research of this nature will be just as important to the future of deep
learning as development of new methods.

We will provide some remarks that we hope will clarify the results for Reviewer
44.
* First, regarding terminology: we will ensure that all terms are clearly
defined before publication. In particular, we use the term "receptive field"
(following other deep learning papers) in analogy with its use in neuroscience
to mean the set of sensory inputs (here, pixels) that influence the output of
a particular node. So, the size of receptive fields of nodes in the conv1
layer is simply the 11x11 filter size, and the receptive fields grow as one
moves to higher layers, eventually encompassing the entire image at the
classification stage.
* In Figure 1, the visual (not semantic) correspondence is precisely what we are
claiming; indeed, the images may be like photomosaics. A priori, given the
extensive internal pooling of "Alexnet", it is not obvious that the activation
features will retrieve patches that are visually similar in the small (16x16
pixel) center regions of their receptive fields. However, this is precisely
what we find, and we conduct the following experiments to test whether this
visual correspondence is semantically valid, at least for part
detection. (We did not notice blurriness when viewing this figure
electronically, but it may not print well; we'll check this before
publication.)
* (Following the definition of receptive field above,) Figure 2 presents
averages in pixel space over patches that are nearby in feature space. The
purpose of this figure is to provide another view on how the features respond
preferentially to the centers of their receptive fields, and to provide
suggestive evidence about the size and shape of the preferred center region.
* Indeed the results in Figure 3 do not show a compelling superiority of either
SIFT or conv4 features, as Table 2 confirms. We present this figure to
illustrate the task and show the (somewhat distinct) types of errors that each
feature makes. We think the on-par performance of convnet features and SIFT
for alignment is a significant result; the layered pooling of the convnet
features has not destroyed the local information needed to align as well as
SIFT, contrary to the intuitions of many.
* We agree that this work suggests replacing SIFT with CNN features; in fact,
the concurrent work cited as [26] does this for descriptor matching, and
includes a computational comparison.

We agree with the comments of Reviewer 8. In particular, we are sorry to give
"conspicuously few hypotheses for the mechanisms why the conv-net features do as
well as they do". Although we would like a clear answer to that question, we
wrote this paper because we think that having extensive knowledge of the
empirical properties of convnet features will be a useful step towards gaining a
complete understanding of how convnets learn.

Overall, we feel that the most important reason this work should be at NIPS is
because, in the words of Reviewer 29, "the results are surprising"! Science is a
process of updating our beliefs, and less expected evidence means bigger
updates.